# Advances in Modeling Approaches for Oral Drug Delivery: Artificial Intelligence, Physiologically-Based Pharmacokinetics, and First-Principles Models

**DOI:** 10.3390/pharmaceutics16080978

**Published:** 2024-07-24

**Authors:** Yehuda Arav

**Affiliations:** Department of Applied Mathematics, Israeli Institute for Biological Research, P.O. Box 19, Ness-Ziona 7410001, Israel; yehuda.arav@gmail.com

**Keywords:** mathematical models, artificial intelligence, machine learning, deep learning, QSAR, PBPK, CFD, molecular dynamics

## Abstract

Oral drug absorption is the primary route for drug administration. However, this process hinges on multiple factors, including the drug’s physicochemical properties, formulation characteristics, and gastrointestinal physiology. Given its intricacy and the exorbitant costs associated with experimentation, the trial-and-error method proves prohibitively expensive. Theoretical models have emerged as a cost-effective alternative by assimilating data from diverse experiments and theoretical considerations. These models fall into three categories: (i) data-driven models, encompassing classical pharmacokinetics, quantitative-structure models (QSAR), and machine/deep learning; (ii) mechanism-based models, which include quasi-equilibrium, steady-state, and physiologically-based pharmacokinetics models; and (iii) first principles models, including molecular dynamics and continuum models. This review provides an overview of recent modeling endeavors across these categories while evaluating their respective advantages and limitations. Additionally, a primer on partial differential equations and their numerical solutions is included in the appendix, recognizing their utility in modeling physiological systems despite their mathematical complexity limiting widespread application in this field.

## 1. Introduction

Oral drug delivery is the most comfortable and commonly utilized route for administering drugs, with about 60% of all dosage forms in the worldwide market being oral [1,2]. It has attracted attention due to its unique advantages, including sustained and controllable delivery, ease of administration, feasibility for solid formulations, patient compliance, and an intensified immune response in the case of vaccines [3].

However, the extent and rate of drug absorption from the gastrointestinal (GI) tract into the bloodstream is a complex process that is influenced by various factors. These can be categorized into three main classes [4] (Figure 1): the first encompasses the physicochemical and biological properties of the drug, including solubility, pKa, chemical and biological stability, and lipophilicity; the second involves the physiological and anatomical properties of the GI tract, such as pH levels, gastric emptying rates, transit times through the small and large bowel, concentrations of micelles, active transport mechanisms, efflux processes, and gut wall metabolism; and the third comprises formulation factors, including dosage form, particle size distribution, and the properties of various delivery systems such as solutions, tablets, capsules, suspensions, emulsions, gels, and modified-release formulations.

Given this complexity, designing an oral formulation capable of achieving an appropriate pharmacokinetic profile for new drugs or optimizing existing drug formulations necessitates a methodology to estimate the extent and dynamics of absorption [5]. However, conducting extensive in vivo evaluations in preclinical species can be both costly and time consuming, and only provides information specific to the conditions under which the experiments were conducted. Moreover, inter-species differences may result in unreliable predictions of oral bioavailability in humans when relying solely on in vivo animal data.

Mathematical models for oral drug absorption have increasingly influenced the development of oral drug delivery products over the past two decades, reducing the failure rate of new drugs due to poor drug-like properties from 30–40% during the 1990s to approximately 10–15% toady [6]. Over the years, various types of mathematical models have been developed, generally falling along a spectrum between data-driven, mechanism-based, and first principles models. In its purest form, a data-driven model utilizes solely the observed data of the phenomena it represents. For example, characterizing the distribution and statistical properties of a measured variable is a data-driven model. Mechanism-based models describe the dynamics of the system by detailing the kinetics of the relevant processes after simplifications. For example, the Michaelis–Menten approximation [7] is often used to describe the metabolism rate of the hepatic enzyme CYP3A4, which is involved in the metabolism of many drugs, rather than detailing its full set of reactions [8]. Conversely, first principles models rely solely on fundamental principles such as thermodynamics, conservation of mass, energy and momentum to describe the system dynamics and estimate the parameters. For instance, the Einstein–Stokes equation can be used to calculate the diffusion coefficient of a dye in still water [9]. Different model types have advantages and disadvantages regarding the questions at hand. This work aims to review the different types and provide insights into their strengths and weaknesses.

## 2. Overview of the Physiology and Mechanisms of Human Oral Drug Absorption

Oral drug delivery commences with the administration of a drug formulation to the buccal cavity (Figure 2). The buccal/sublingual route is an attractive route of delivery for some highly water-soluble and absorbable drugs [10] due to its high compliance, avoidance of the hepatic first-pass metabolism, and quick onset [11]. In the following, we provide an overview of absorption in the buccal/sublingual route; for further details, see [12,13,14,15].

Following the disintegration of the buccal/sublingual formulation, the drug has to dissolve in the available limited volume of human saliva (0.8 to 1.1 mL, [11]) and permeate the epithelium. The buccal epithelium has a surface area of approximately 50 cm^2^ [13] and consists of 40–50 layers, only a few of which are shown in the schematics that describe the buccal/sublingual absorption route in Figure 2.

The permeation of drugs across the epithelium and into the bloodstream occurs through various mechanisms, which involve either passive or active transport [16]. Passive transport is facilitated by the concentration gradient across the membrane, and can occur through either the cell membrane (transcellular transport) or through the paracellular pathway (in which the drug traverses intercellular spaces). The paracellular route primarily facilitates the absorption of hydrophilic drugs, while transcellular transport is predominant for lipophilic compounds [15]. Additionally, active transport mechanisms such as carrier-mediated transport contribute to the transportation of certain drugs across the oral mucosa [13,17].

Several observations (see Vondracek et al. [18] and reference therein) support the presence of the cytochrome P450 (CYP) family in the buccal epithelium. The CYP family of enzymes is responsible for the oxidative metabolism of numerous endogenous and exogenous chemicals [18]. Consequently, certain drugs that permeate the epithelium via the transcellular route may undergo metabolism, potentially reducing their bioavailability.

The buccal/sublingual route might be less preferable for drugs with poor water solubility, which comprise approximately 40% of the drugs on the market. The low solubility of these drugs combined with their relatively small surface area and short residence time due to accidental swallowing and dilution with saliva [19] can limit their absorption. To increase the relevancy of this route, there is an ongoing effort to develop novel delivery methods that can increase drug solubility [11,20,21,22,23] or increase residence time by developing formulations based on bioadhesive polymers [19].

Hence, the majority of formulations are intended for absorption in the gastrointestinal tract (GIT) and reach the stomach after ingestion (Figure 2). The stomach, resembling a bean-shaped muscular bag, serves as both a reservoir and a regulator for transfer to the small intestine (SI) via the pylorus sphincter [24]. Physiological conditions within the stomach vary depending on its contents [25]. In fasted state, the stomach typically maintains a pH of approximately 1–3 [3,25], possesses a surface area of roughly 3.5 m^2^ [1], and contains 200–250 mL of luminal fluids [25,26] containing specific enzymes crucial for food digestion [16]. Under these conditions, solid dosage forms such as tablets, capsules, and pellets may remain in the stomach for up to 2 h [27], while the emptying rate of liquid solutions is proportional to the stomach content volume (e.g., first order), with a half-emptying time ($t_{1/2}$) typically ranging from 5 to 20 min [26,28].

After a meal, the pH of the stomach increases to between 1.88 and 4.98 and its volume increases to 500 mL or higher [25]. Under these conditions, the residence time of solid dosage can be 3–5 h [27], while the emptying rate of fluids becomes constant (e.g., zero order) and may last approximately 120 min [26] depending on the caloric intake (low-fat or high-fat) of the meal [25,26]. For further information on the stomach’s anatomy and physiology in the presence or absence of food, see Cheng et al. [25]. The drug is emptied from the stomach into the small intestine (SI).

The SI is a lengthy (280 cm) and narrow tube (≈1–2 cm diameter, [24]) that is primarily responsible for drug and nutrient absorption [24,29]. It connects to the stomach via the pylorus sphincter at one end and the large intestine through the ileocecal sphincter at the other. Anatomically, the SI is segmented into three distinct regions: the duodenum (20 cm long), the jejunum (104 cm long), and the ileum (156 cm long) [24]. Surface area amplification during intestinal fluid contact occurs through three structures: large folds, finger-like projections (villi), and smaller protrusions (microvilli) [24,29]. Large folds are typically located between the mid-duodenum and mid-ileum, with their size and density decreasing along the SI. Similarly, the amplification of villi and microvilli diminishes along the length of the SI [24,30]. This results in a surface area of approximately 250–300 m^2^ [3,24]. The pH increases gradually along the SI, from 5.5 in the duodenum to 7–9 in the distal ileum [3,25]. The contents of the SI are propelled toward the ileocecal valve through contraction of its walls. The transit time is 3–4 h, independent of the presence of food and the form of the pharmaceutical formulation (tablets, pellets, and liquids) [27].

The large intestine (LI) resembles an imperfect cylinder approximately 1.5 m long and with a diameter ranging from 4 to 8 cm [3,31], with a surface area of approximately 1.3 m^2^ [1]. Its roles encompass the absorption of electrolytes and water, fermentation of unused energy substrates, immune system priming, production and absorption of vitamins and hormones, fecal transport to the rectum, and fecal storage until elimination [31]. Anatomically, the LI is segmented into four major parts: the ascending colon or cecum (20–25 cm), the transverse colon (40–45 cm), the descending colon (10–15 cm), and the sigmoid colon (35–40 cm) [31]. The pH within the colon varies slightly across different regions: 6.2–7.4 in the cecum, 5–8 in the transverse colon, 6–8 in the descending colon, and 7–8 in the sigmoid colon [31]. Undigested food is transferred from the small intestine to the cecum, resulting in initially liquid feces that gradually solidify as they traverse the LI [31]. Transit time through the LI is highly variable, influenced by factors such as diet, stress, mobility, medication, illness, and gender [32], and is estimated to range between 5 to 73 h [33].

A solid formulation releases the drug as it progresses through the gastrointestinal tract (GIT). The rate of drug release from a formulation and its dissolution along the GIT depend on several factors, including the properties of the formulation itself, the physicochemical properties of the drug, and the composition of the GIT fluids. Formulations are frequently categorized according to the pattern of drug release from tablets, which commonly includes immediate-release, modified-release, and delayed-release formulations [34]. Immediate-release formulations rapidly release the drug after administration, and represent the most prevalent type of tablet. Examples include disintegrating, chewable, effervescent, sublingual, and buccal tablets [34]. Modified-release formulations are designed to release the drug over time. Finally, in delayed-release formulations the drug is liberated from the tablet sometime after administration, often to protect the drug from the gastric environment [34]; see Taylor and Aulton [34] and Alqahtani et al. [1] for an extensive review of oral drug formulations and Homayun et al. [3] for a review of advances and upcoming technologies in oral administration.

When the drug that was released from the formulation is solid, it must dissolve in the GIT fluids before it can permeate the intestinal epithelium and reach the blood. The dissolution rate of the drug depends on the powder distribution size [30,35], the charge of the drug, and the pH of the environment in which it dissolves [5]. Following food intake, bile secretions from the gall bladder aid in the dissolution and permeation of lipophilic compounds [36].

After dissolution, the drug becomes susceptible to chemical and biological degradation within the lumen of the GIT (Figure 2, Gastrointestinal route). Chemical degradation can occur under fasted conditions post-administration due to the harsh acidic environment within the stomach [3]. Biological degradation may occur in the stomach through gastric enzymes such as pepsin and gelatinase as well as in the upper SI via digestive enzymes secreted by the pancreas, including lipases (fat degradation), amylase (starch degradation), peptidases (peptide disintegration), and trypsin (protein decomposition) [3]. Additionally, brush-border metabolism occurs on the surface of the SI by enzymes present within the brush-border membrane, such as isomaltase, alkaline phosphatase, sucrose, and other peptidases [1]. Further degradation may arise from the intestinal flora of the colon, which are primarily located in the lower portion of the GI tract [1].

The permeation of free drugs through the gastrointestinal (GIT) epithelium occurs through mechanisms similar to those described for the buccal/sublingual absorption route. Specifically, drugs are absorbed either passively (via transcellular or paracellular routes) or through active transporters [3,4]. Unlike the buccal/sublingual epithelium, the GIT epithelium is arranged in a single-column layer (monolayer) primarily composed of enterocytes [1]. These cells express efflux transporters from the ATP-binding cassette (ABC) superfamily, along with cytochrome P450 (CYP) enzymes responsible for intracellular metabolism.

ABC transporters, including P-glycoprotein (P-gp, MDR1, ABCB1), multi-drug resistance-associated proteins (MRPs), and breast cancer resistance protein (BCRP, ABCG2), function to limit the intracellular accumulation of their substrates by facilitating efflux out of cells [37]. These efflux transporters, along with CYP enzymes, exhibit overlapping substrate specificity. Their combined action prolongs drug exposure to CYP enzymes while maintaining low concentrations, thereby preventing saturation [1,38]. This mechanism can limit the bioavailability of many drugs, and may lead to drug–drug interactions, as some drugs can inhibit either P-gp or CYP enzymes [1]; see Murakami et al. [37] for an extensive review of efflux proteins in the intestine and Alqahtani et al. [1] for further discussion on the metabolic barriers to absorption.

Following absorption in the GIT, the drug reaches the portal vein and passes through the liver before it reaches systemic circulation [1,38,39,40,41]. As the drug passes through the liver, it might be metabolized by the hepatocytes of the liver that express the CYP, as well as by other enzymes (first-pass metabolism). This first-pass metabolism represents a major metabolic barrier to drugs administrated through the gastrointestinal route [1].

## 3. Approaches for Mathematical Modeling

Mathematical modeling and computer simulations have become increasingly valuable in facilitating different aspects of the development of oral delivery of drugs. Due to the complexity of the oral absorption process and its many aspects, different models have been developed with various modeling techniques.

Generally, these models can be categorized into three different approaches: data-driven models, mechanism-based models, and first principles models (see Figure 3).

Data-driven models are developed based on empirical data and do not rely on explicit theoretical principles. These models extract patterns and trends from observed data, and are valuable for summarizing and presenting data concisely. Additionally, they can make predictions within the range of observed data, provided a sufficient amount of data is available. Hence, the utility of these models is limited to scenarios where ample empirical data are available. For example, Fagerholm et al. integrated fourteen different data-driven models to predict various pharmacokinetic parameters and used them to forecast the oral bioavailability of drugs [42].

First principles models leverage fundamental physical principles, including the conservation of mass, momentum, and energy as well as principles from thermodynamics, fluid dynamics, and other relevant disciplines. These models describe system dynamics by considering factors such as geometry, boundary conditions, and external forces. They provide detailed insights into phenomena and facilitate the investigation of dynamics across a wide range of parameter values. For example, Palmada et al. [43] used computational fluid dynamics (CFD) to investigate the effect of motility patterns on the flow field in the duodenum. However, due to their inherent complexity, first principles models require a wealth of detailed information, often limiting their applicability to relatively small physical domains.

Mechanism-based models integrate both approaches by representing system dynamics with simplified versions of first principles while incorporating data-driven models to address the complexities of the processes that were simplified. For instance, Physiologically-Based Pharmacokinetics (PBPK) models describe the pharmacokinetics of drugs in the body by treating each organ as a well-mixed component. These models simplify Fick’s law of diffusion to account for drug uptake into organs, with the rate constants being derived from experiments on drug biodistribution [44,45].

In the following section, we examine the application of models employing various approaches for oral drug absorption.

## 4. Data-Driven Models

Data-driven models, also known as empirical models, are utilized to quantitatively describe empirical datasets and predict various properties. In the context of oral drug absorption, they can generally be categorized into three subcategories: conventional pharmacokinetics models, conventional quantitative structure–activity relationship (QSAR) models, and artificial intelligence (AI) models that utilize machine and deep learning models (ML and DL, respectively), also referred to as deep QSAR [46], or nonlinear QSAR [46].

In the following sections, we review the advancements made in each subcategory regarding oral drug absorption. For a comprehensive overview of classical pharmacokinetics models, refer to [47]. To explore QSAR models in detail, see [48,49]. Additionally, for insights into the applications of AI in pharmacy, consult [50].

### 4.1. Conventional Pharmacokinetics Models

Conventional pharmacokinetics models most commonly describe the kinetics of the drug in the plasma following oral absorption by fitting the plasma concentration profile in time to a two-exponent function [40,47]:(1)$$C_{p}=\frac{F\cdot k_{a}\cdot D}{V(k_{a}-\frac{CL}{V})}\cdot \left(e^{-\frac{CL}{V}\cdot t}-e^{-k_{a}\cdot t}\right)$$
where $C_{p}$ is the concentration in plasma, *F* is the bioavailability, $k_{a}$ is the absorption rate coefficient, *D* is the administrated dose, *V* is the volume of distribution of the drug in the body, and CL is the clearance.

The parameters of Equation (Equation 1) usually cannot be estimated from the data of a single oral dose [40]. Hence, additional data are needed to estimate other parameters. The values for *F*, CL, and *V* are usually computed from plasma concentrations following intravenous (IV) administration. Under certain circumstances, estimation of *F* can also be achieved from urine data [40]. Due to the importance of these pharmacokinetic parameters, several studies have presented methods for assessing them using QSAR; related ML and DL models are reviewed in Section 4.2 and Section 4.3, respectively.

Equation (Equation 1) is derived by modeling the absorption from the gastrointestinal tract (GIT), as depicted in Figure 2 (gastrointestinal absorption route), as a first-order process with a rate constant $k_{a}$. In this model, the body is treated as a well-mixed compartment and elimination processes from the body are assumed to be first-order. Therefore, Equation (Equation 1) represents the solution of the ordinary differential equations (ODE) system [47]:(2)$$\begin{array}{cc} \frac{dA_{1}}{dt} & =-k_{a}\cdot A_{1} \end{array}$$(3)$$\begin{array}{cc} V\frac{dC_{p}}{dt} & =\underset{\mathrm{Absorption}\;\mathrm{rate}}{\underbrace{k_{a}\cdot A_{1}}}-CL\cdot C_{p} \end{array}$$(4)$$\begin{array}{cc} A_{1}\left(0\right)=F\cdot D & \;\;\;\;;\;\;\;\;C_{p}\left(0\right)=0 \end{array}$$
where $A_{1}$ is the amount of drug in the GIT.

The value of $k_{a}$ is determined by fitting empirical data. Consequently, Equation (Equation 1) allows for the quantitative comparison of two immediate-release formulations when the plasma concentration time series is acquired. However, the assumption of first-order absorption usually implies that the rate-limiting process in absorption is the permeation through the intestinal epithelium and that the absorption mechanism is passive; therefore, this model is limited in its applicability to soluble drugs administered in a fasted state within immediate-release formulations. Additionally, because $k_{a}$ is derived from the plasma concentration time series, estimating its value before experimentation data are available is challenging. To address these limitations, more sophisticated models that incorporate additional details of the absorption mechanisms are needed. These advanced models are discussed in Section 5.

### 4.2. Conventional Quantitative Structure–Activity Relationship (QSAR)

Conventional QSAR models use multivariate statistics to establish a linear relationship between values of chemical descriptors computed from molecular structures and experimentally measured chemical properties such as charge, solubility, and partition coefficient along with bioactivities such as passive intestinal permeability and intrinsic hepatic clearance [48,51]. The assumption that underlies conventional (linear) QSAR modeling is that similar compounds exhibit similar biological effects and that gradual changes in compound structure are accompanied by gradual changes in potency, the so-called similarity–property principle (SPP) [48]. See [46,48] for a review of QSAR modeling.

Estimating the absorption, distribution, metabolism, and elimination (ADME) of a new compound is an essential step in the screening process [16,49]. Over the past two decades, numerous QSAR models have been developed to predict these properties. For a comprehensive review of QSAR models for predicting ADME properties, refer to [49,52,53,54]. Here, we provide an short overview.

Estimating the oral absorption is a key step in screening new drug candidates. The drug solubility and permeability in the GIT together with the hepatic first pass are considered to be the leading factors that determine *F* [52]. Hence, QSAR models have been developed to estimate to predict the permeability [52,55,56,57,58], solubility [42,55,59], and intrinsic hepatic clearance [42] and to predict *F* either directly from molecular descriptors [42,55,60,61,62,63,64] or by using the output of other QSAR models [65]. Because efflux transporters such as P-gp may play an important role in the absorption, QSAR models have been developed to estimate the affinity of the drugs to P-gp [66]. However, training a QSAR model to predict *F* directly from the drug descriptors without explicitly considering the properties of formulation (e.g., particle size, disintegration rate, and dissolution rate) might lead to inconsistencies. For example, the bioavailability of griseofulvin is doubled following administration as a micronized formulation [30]. Hence, it is important to perform rigorous quality checks on the database being used and ensure that all the entries describe similar absorption conditions. QSAR models for estimating drug distribution include the prediction of *V* [67] and its protein binding [42,68,69,70]. QSAR models that estimate the first-pass metabolism and elimination can predict the intrinsic hepatic clearance [71,72,73] and the systemic clearance [72,73]. QSAR models have also been developed to estimate the buccal permeability [74] and bioavailability [75].

QSAR modeling has been utilized in the design of formulations, as mixing the drug with various excipients to create a formulation can enhance its solubility, efficacy, and stability. The selection of suitable excipients depends on the physical and chemical properties of the drug as well as on the type of formulation [76]. Specifically, QSAR models have been developed for selecting excipients [77], ensuring formulation stability [78], and designing controlled-release formulations [79]; see Kulkarni et al. [80] and Aguillar et al. [81] for a review on QSAR modeling in formulation design.

QSAR models that predict a numerical property value based on the molecular characteristics of a drug are often preferred, as they allow for molecule selection based on arbitrary criteria. However, achieving an adequately accurate regression model can often prove challenging, and classification may come in handy. Classification models determine whether a molecule belongs to a specific property class (e.g., ‘high’ or ‘low’), sometimes providing the probability of class membership as well [60,82].

### 4.3. Artificial Intelligence (AI)

AI models leverage ML and DL algorithms to augment traditional QSAR models and perform large-scale dataset classification. These models are categorized as supervised and unsupervised learners based on their learning processes (Figure 4). Supervised learning employs established input–output relationships, with known features utilized to predict output values, while unsupervised learning focuses on discovering hidden structures within data, aiming to group similar instances together based solely on their inherent characteristics [50,83].

Supervised learning algorithms enhance traditional QSAR models by establishing nonlinear relationships between chemical descriptors derived from molecular structures and experimentally measured chemical and biological activity properties [48,50]. The transition from linear to nonlinear relationships is pivotal, particularly as databases expand to include structurally diverse active compounds with few or no common scaffolds. In such cases, departure from the SPP assumption becomes crucial. Structural differences among active compounds are often abrupt and non-gradual, necessitating the adoption of nonlinear modeling approaches in order to satisfactorily capture the structure–activity relationship (SAR) [48]; see Muratov et al. [48] for a review of different supervised learning algorithms used in QSAR modeling and Sarkar et al. [83] for a review of AI algorithms.

In recent years, nonlinear QSAR models based on ML and DL techniques have become the most popular strategy to develop QSAR models for prediction of the ADME properties of drug candidates [84] and formulation design [85]. Similar to conventional QSAR, nonlinear QSAR models can be developed to estimate the permeability [49,82,86], physio-chemical properties [58,59], distribution [49,82], affinity to P-gp [87,88], hepatic clearance [49,82], metabolism by the CYP family [89,90,91,92], and *F* in the fasted [59] and fed [93] states. For reviews on using AI for ADME properties and other uses in the pharmaceutical industry, see [46,49,50,53,94].

Unsupervised learning is a type of machine learning in which the algorithm is not provided with labeled data; instead, it is tasked with identifying patterns and relationships within the data independently. Hence, unsupervised learning algorithms are used to develop clustering methods, which can help in identifying groups and increase the screening efficiency. Butina et al. [95] developed an unsupervised clustering method for the generation of drug groups based on the standard Daylight fingerprints and Tanimoto similarity. Korolev et al. [96] used unsupervised learning to assess the metabolism and toxicity of drugs using a database of known human cytochrome P-450 substrates. Further applications of unsupervised models include reducing the dimensionality of complex higher-dimensional datasets [50,97].

## 5. Mechanism-Based Models

Mechanism-based models combine physiological and physiochemical principles with a simplified representation of underlying mechanisms. Developing such models presents a challenge due to the system’s complexity, and involves decisions about which mechanisms to include and how to determine the appropriate level of detail needed for the representation.

Building on the work of Yu et al. [98], we categorize mechanism-based models into three subcategories: quasi-equilibrium, steady-state, and dynamic or physiologically-based pharmacokinetics (PBPK) models. These distinctions are based on their treatment of spatial and temporal variables [4]. Specifically, Quasi-equalibrium models are independent of spatial or temporal variables, steady-state models are independent of the temporal variables but depend on the spatial variable, and dynamic models depend on both spatial and temporal variables.

### 5.1. Quasi-Equalibrium

One of the earliest theories devised to elucidate the oral absorption process is the pH partition hypothesis [98], rooted in the theoretical framework proposed by Jacobs [99] and extensive studies conducted during the 1950s and 1960s [100,101,102]. According to this hypothesis, only the non-ionic form of the drug can traverse the passive transcellular pathway.

The pH partition hypothesis was subsequently incorporated into the absorption potential (AP) concept introduced by Dressman et al. [103]. This concept qualitatively correlates the extent of drug absorption in the gastrointestinal tract (GIT) to physiological and physiochemical factors such as the administered dose, solubility, lipophilicity, and charge. Mathematically, the AP is defined as
(5)$$AP=\log\left(\frac{PI_{un}}{D_{0}}\right),$$
where AP represents the absorption potential, *P* denotes the partition coefficient, $I_{un}$ indicates the fraction of the unionized form at pH 6.5, and $D_{0}$ is the dose number, provided by
(6)$$D_{0}=\frac{D}{V_{SI}C_{s}}.$$
In this expression, *D* signifies the dose, $C_{s}$ denotes the drug solubility, and $V_{SI}$ represents the volume in the intestine, typically assumed to be 250 mL [103].

Macheras and Symillides [104] used the AP to provide quantitative estimations of *F* for drugs that are not limited by the dissolution rate.
(7)$$F=\frac{{\left(10^{AP}\right)}^{2}}{{\left(10^{AP}\right)}^{2}+I_{un}\left(1-I_{un}\right)}$$
With the constraints that AP=1000 when AP>1000 and that $D_{0}=1$ when $D_{0}>1$ [98].

At first glance, the AP absorption model presented in Equation (Equation 7) appears similar to a QSAR model, utilizing the drug’s physicochemical properties, including charge, lipophilicity, solubility, and administered dose, to predict oral absorption. However, it diverges from QSAR models by integrating a deeper understanding of the physiological and chemical aspects of the absorption process. The assumption that underlies the development of QSAR models is that these connections will emerge given a sufficient amount of data.

### 5.2. Steady-State

Steady-state models were developed to enhance the outcomes of quasi-equilibrium models by adopting a more physiological approach and calculating *F* by considering both dissolution and permeation rates along the small intestine (SI) [98].

Several studies [105,106,107] have employed the macroscopic mass balance approach to calculate F [98]. In this approach, the SI is described as a cylindrical tube with radius $R_{SI}$, and length $L_{SI}$. The absorbed fraction is computed by assuming a steady state and integrating along the SI. This approach has been extended to account for the effect of the drug’s maximal solubility [106] and degradation along the SI [107].

Models developed using the macroscopic approach take into account the solubility limit of the drug, but do not consider its dissolution rate [108]. Oh et al. [109] employed a microscopic approach to develop a model that takes into account the dissolution kinetics, albeit of a monodisperse powder. This model also conceptualizes the SI as a straight cylinder, but assumes that the contents of the SI are convected as a plug flow with a volume of 250 mL, that is, the model neglects the mixing along the SI. Using nondimensional analysis, three dimensionless groups are defined: the dose number ($D_{0}$), the absorption number (An), and the dissolution number (Dn), expressed as follows.
(8)$$\begin{array}{cc} An & =P_{eff}\cdot \frac{\pi R_{SI}L_{SI}}{Q_{SI}} \end{array}$$
(9)$$\begin{array}{cc} Dn & =\frac{DC_{s}}{\rho r_{0}^{2}}\cdot \frac{\pi R_{SI}^{2}L_{SI}}{Q_{SI}} \end{array}$$
Here, $P_{eff}$ is the effective permeability coefficient, $R_{SI}$ and $L_{SI}$ represent the radius and length of the SI, respectively, $Q_{SI}$ denotes the flow in the SI, ρ is the drug density, and $r_{0}$ stands for the initial particle radius. When $D_{0}$ is very high, the drug is not expected to have dissolution problems, while high An reflects a drug with high permeability and high Dn reflects a drug with a high dissolution rate.

These three dimensionless groups formed the basis for the definition of the well-known Biopharmaceutics Classification System (BCS) using An and Dn, as $D_{0}$ was recognized as a factor with lesser importance [28]. In this system, drugs are divided into four classes: Class 1 with high An and high Dn (high permeability/high dissolution); Class 2 with high An and low Dn, provided that $D_{0}$ is not too large (high permeability/low dissolution); Class 3 with low An and high Dn (low permeability/high dissolution); and Class 4 with low An and low Dn (low permeability/low dissolution). The BCS has further adapted to the Developability Classification System (DCS) and the refined Developability Classification System (rDCS), which are tools for assessing the developability of new drug candidates [110]. Wu and Bennet [111] extended the BCS to the Biopharmaceutics Drug Disposition Classification System (BDDCS), which also takes into account the drug’s disposition, including routes of drug elimination and the effects of efflux and absorptive transporters.

### 5.3. Dynamic Physiologically-Based Pharmacokinetics (PBPK) Models

Dynamic Physiologically-Based Pharmacokinetics (PBPK) models were developed to calculate the absorption rate by incorporating the kinetics of physiological processes in oral drug absorption. These models are often integrated with pharmacokinetics models to predict plasma concentration time profiles.

Dynamic PBPK models for oral absorption consider both spatial and temporal variables. Generally, these models are classified into compartmental and dispersion models. While both types of models treat the temporal variable as continuous, they differ in their treatment of the spatial variable; compartmental models represent the spatial variable as a series of one or more well-mixed compartments, whereas dispersion models treat it as a continuous variable.

In the following section, we review these two subtypes.

#### 5.3.1. Compartmental Models

Compartmental models represent the GIT as a series of one or more well-mixed compartments, with each compartment maintaining a uniform concentration. The transfer kinetics between these compartments are assumed to be linear. Representing the SI as a series of well-mixed components is a rough approximation of the spatial variable that has no physical basis [108]. As such, the number of compartments varies between one and four compartments [26,112,113,114,115,116,117,118]. However, because the number of compartments affects the simulation results, Yu et al. [108] set the number of compartments to 7 by fitting the model prediction of the SI intestinal transit to the mean measured time [108]. The compartments were then correlated to the physiology based on transit times [98,108]. The first half of the first compartment represents the duodenum, the second half of the first compartment along with the second and third compartments represents the jejunum, and the remaining compartments represent the ileum.

The seven-compartment intestinal transit model was later extended to account for the absorption of a passive fully dissolved drug in the compartmental absorption and transit (CAT) model [119]. In this approach, the intestinal transit and absorption are described with the following system of ODEs:(10)$$\frac{dY_{n}}{dt}=K_{t}Y_{n-1}-K_{t}Y_{n}-k_{a}Y_{n}\;\;\;\;\;\;\;n=1..N$$
where N=7 is the number of compartments, $Y_{n}$ is the fraction amount of drug in compartment *n*, and $K_{t}$ is the transfer rate constant.

The Advanced CAT (ACAT) [41] and Advanced Dissolution, Absorption, and Metabolism (ADAM) [120,121] models extended the CAT model to include first-pass metabolism, absorption from the LI, dissolution and solubility, release kinetics, intestinal metabolism, active transport, and food effects. The two models differ in their modeling approach to the dissolution process. The ACAT model uses a Noyes–Whitney approach that utilizes the film model [122], while the ADAM model corrects for the spherical shape of the particle [123].

The commercially available version of the ACAT model, GastroPlus^TM^, includes several additional improvements that have facilitated its use. It has been used to justify a biowaiver for a selected BCS 2 compounds [124] and to determine the impact of different formulation factors on the absorption of drugs, including solubility, particle size, size distribution, and presence of food [125,126,127,128,129,130,131]. The ADAM model is also available commercially as Symcyp^®^, and has been used to predict the effect of the drug release rate on its absorption [120,132,133,134], to optimize drugs for pediatric use [135,136], and to investigate drug–drug interactions [137]. Additional compartmental models are reviewed in Huang et al. [4].

Compartmental models have gained much popularity in estimating oral drug absorption thanks to their mathematical simplicity, intuitive nature, and easy correlation with pharmacokinetics models as well as to the availability of commercial software.

However, correlating a compartment with a physiological segment of the SI lacks a physical basis [108]. To delve deeper into this issue, we must consider that intestinal motility both propels and spreads intestinal content along the SI [108]. Consequently, as each compartment is well-mixed, the number of compartments *N* dictates the extent of ‘spread’, while the kinetic transfer rate $K_{t}$ governs the mean velocity (Figure 5a). Therefore, determining *N* based on the mean transit time of the SI, as in the CAT model procedure [108], does not guarantee that the model will accurately reflect the ‘mixing’ rate observed in the SI, as it fails to account for this factor. As *N* approaches infinity, the model converges to a dispersion model with only the convection term and without the mixing term; refer to Section 5.3.2 and Appendix A for further details.

Consequently, the determination of the volume available for drug dissolution lacks a robust physiological foundation, potentially leading to underestimation or overestimation of the actual surface area available for absorption, and consequently of the drug concentration. Moreover, this discrepancy may result in overestimation or underestimation of the efflux transport and enzymatic metabolism, as these processes are contingent upon the drug’s concentration.

Another implication of using the compartmental approach pertains to the description of the dissolution rate. The dissolution rate of a drug from a powder particle, whether monodisperse or polydisperse, depends on the amount of the drug and the particle radius [122,123]. However, in compartmental models the kinetics of the undissolved drug are simplified and track only a single variable; consequently, to account for dissolution, the particle radius must remain fixed, as only the amount of undissolved drug is computed. This could potentially lead to an underestimation of the dissolution rate, as the dissolution rate increases with decreasing particle size due to the increase in surface area [30].

As a result of these simplifications, compartmental models might overestimate or underestimate absorption. Arav et al. [30] demonstrated that the noncommercial version of the ACAT model overestimated the absorption of griseofulvin in both micronized and non-micronized formulations. The continuous models described below address some of these issues.

#### 5.3.2. Continuous Models

The continuous model was first suggested by Ho et al. [138], and describes the SI as a straight tube with length $L_{SI}$ and radius $R_{SI}$. The concentration in the tube C(x,t) is assumed to be well-mixed in the radial direction; therefore, the concentration depends only on the distance *x* from the pylorus and the time. Intestinal motility is described as a convection with uniform velocity and dispersion that accounts for the mixing (Figure 5b). The equation that describes the dynamics of intestinal motility and the passive absorption of a completely dissolved drug is the following partial differential equation (PDE):(11)$$\frac{\partial C}{\partial t}+\frac{\partial }{\partial x}\underset{SI\;motility}{\underbrace{\left[\underset{Convection}{\underbrace{uC}}-\underset{Mixing}{\underbrace{D_{eff}\frac{\partial }{\partial x}C}}\right]}}=\underset{\mathrm{Absorption}}{\underbrace{-\frac{2\cdot SA_{SI}\cdot P_{int}}{R_{SI}}C}}$$
where *u* is the axial velocity, $D_{eff}$ is the mixing rate (dispersion) coefficient, $R_{SI}$ is the intestinal radius, $SA_{SI}$ is the surface area amplification, and $P_{int}$ is the intrinsic permeation coefficient of the drug [30,98]. The influx from the stomach and the outflux to the LI are provided by the boundary conditions (see Appendix A for further details).

While the continuous model provides a more physiologically accurate representation of the small intestine (SI) [98], its mathematical complexity has hindered its widespread adoption; since its introduction in 1983 by Ho et al. [138], only a few studies have employed it [24,30,139,140].

The mathematical complexity of Equation (Equation 11) entails two main challenges: first, understanding the interpretation of the equation, and second, devising methods to solve it. We provide a detailed explanation of these two aspects in Appendix A; here, we provide a brief overview.

The left-hand side (LHS) of Equation (Equation 11) delineates the SI motility, while the right-hand side (RHS) denotes the alteration in concentration due to absorption. The crucial distinction lies in the fact that the LHS terms merely redistribute the drug along the SI, thereby maintaining the total amount as a constant, whereas the RHS describes a process that diminishes the total amount (e.g., a sink term). SI motility is described as a combination of two processes. The uniform movement along the SI is delineated by a convection term, while the mixing along the SI is described as a dispersion term, similar to the transfer of material by turbulence eddies in highly turbulent regimes [141]. The dispersion term describes mixing, as it corresponds to a flux proportional to the concentration gradient moving from a region of high concentration to a region of low concentration. It is worth noting that $u,\;D_{eff},\;SA_{SI}$, and $P_{int}$ can depend on either the spatial variable, the temporal variable, or both.

Solving the model of Equation (Equation 11) can be achieved with an analytical solution for simplistic cases. Ho et al. [138] used it for the case of the stomach as an infinite reservoir and took into account the passive absorption of completely soluble drug. Wilmann et al. [24,139] extended this model to account for stomach emptying and took into account the limited solubility of the drug assuming a rapid dissolution rate.

However, for more detailed equations the models have to be solved numerically. Arav et al. [30] extended the model to account for the dissolution rate of polydisperse powders in the stomach and SI, and developed computer software to solve this nonlinear set of PDEs. The model of Arav et al. [30] addresses two key limitations of the compartmental model: first, it provides a more physiologically accurate representation of mixing, and second, it considers changes in particle size distribution resulting from dissolution. The model was then used to find the optimal particle size distribution (mean and geometric standard deviation) ensuring that dissolution is not the rate-limiting step in absorption. However, the mathematical aspects of this model are somewhat intricate, and are further elucidated in Appendix B.

Recently, the dispersion model was further developed to elucidate the absorption kinetics of levodopa following controlled-release administration [140]. The findings refute the previous hypothesis of an ‘absorption window’ in the upper small intestine for levodopa [142], demonstrating instead that levodopa is uniformly absorbed along the entire small intestine. Subsequently, the model was leveraged to optimize the properties of the controlled-release formulation to prolong therapeutic efficacy.

## 6. First Principles Models

First principles models utilize fundamental physical and chemical principles to elucidate phenomena in great detail. Compared to mechanism-based models, they provide a more intricate description of the system, enabling exploration of aspects where experimental data may be lacking or difficult to obtain. For instance, Lee et al. [143] investigated the impact of posture and gastroparesis on drug dissolution and distribution within the stomach using a biomimetic in silico simulator based on realistic stomach anatomy and morphology. However, owing to their heightened complexity, the application of these models is typically constrained to specific temporal and spatial domains, focusing solely on particular aspects of the overall oral drug absorption process.

In the context of oral drug absorption, we can distinguish between molecular modeling, which simulates biomolecular dynamics at an atomistic level, and continuum models, which describe the movement and behavior of drug substances within the gastrointestinal (GI) tract as continuous entities without considering individual particles or molecules.

### 6.1. Molecular Modeling

Molecular models simulate biomolecular dynamics at an atomistic level [144,145]. This field encompasses various applications, with molecular docking and molecular dynamics (MD) simulations being widely utilized components within the pharmaceutical domain. These computational approaches play a crucial role in identifying potential leads for subsequent in vitro and in vivo experimental testing [146].

MD simulation predicts the dynamics of the atoms’ position at a molecular level. Similarly to docking simulations, MD simulations are based on the structure of the biomolecules. In traditional “all-atom” MD simulations, the model includes all the interacting atoms in the system, including the solutes. Due to this high level of detail, such simulations are limited in both the length and time scales. Currently, simulations include $10^{8}$–$10^{9}$ atoms and run to μs-scale [145]. In the context of oral drug absorption, MD is applied to estimate drug permeability and solubility. Unlike data-driven models, MD can provide some mechanistic understanding that can be used for optimization [147]. Other uses include optimization of nanoparticle design and function as well as drug loading release profiles [147]. For recent reviews on the applications of MD in drug discovery, see [145,148,149,150,151,152,153,154]. Recent reviews can be found on drug formulation and drug delivery aspects as well [147,155,156,157].

### 6.2. Continuum Models

Continuum models encompass various phenomena as continuous fields, including mass transfer, fluid flow, and elastic deformations. This implies that the model’s variables are typically functions of spatial and temporal variables, akin to the continuous models represented in Equation (Equation 11). The distinction between mechanism-based models and first principles continuum models lies in the level of simplification. For instance, the mechanism-based dispersion model encapsulates fundamental mechanisms of intestinal motility such as convection and longitudinal spreading as uniform flow and longitudinal mixing, with the parameters being derived from experimentation [30]. In contrast, the first principles-based continuum model devised by Fullard et al. [158] delineates intestinal motility by computing the elastic deformation of the intestinal walls coupled with the incompressible Navier–Stokes equations to predict the flow patterns in a rabbit intestine as a result of the contractions.

Hence, continuum models prove to be particularly valuable when tackling problems for which acquiring adequate experimental data is challenging but where a thorough comprehension of the physical system is available. Several studies have recently addressed gastric and intestinal motility [43,143,158], dissolution [159], drug permeation [160], and the design of controlled-release formulations [161].

## 7. Discussion

Oral drug delivery is favored as a route for administering drugs due to its non-invasive nature, convenience, and high patient compliance [1,2]. However, the physiological, physical, and chemical intricacies of the oral route pose challenges in developing new formulations and ensuring the acceptable bioavailability of new drug candidates. Because the experimental trial-and-error approach is expensive, mathematical models have been developed over the past few decades as valuable tools for integrating experimental data, reducing the number of required experiments [162].

Mathematical models are designed to address a certain problem by taking the data that the modeler deemed relevant as input and then producing the desired output. Over the years, different approaches have been taken to developing data-driven, mechanism-based, and first principles-based models (Table 1). Each approach addresses a different problem domain.

Data-driven models, including conventional PK, QSAR, ML, and DL models, are essentially ‘black-box’ systems that correlate specific inputs to outputs. Consequently, these models require a substantial dataset, and operate based on the assumption that the information contained within the data is sufficient for classifying or predicting new data. Essentially, they interpolate data from the provided dataset, making them suitable for high-throughput screening; however, due to their ‘black-box’ nature, understanding the reasons for mispredictions and finding ways to correct them can be challenging.

Mechanism-based models, including quasi-equilibrium, steady-state, and PBPK models, incorporate physical and chemical principles while simplifying the dynamics of various physiological processes. Developing a mechanism-based model involves the modeler’s decision as to the importance of different processes and their mathematical description, often relying on in vivo or in vitro experimental data as well as other mathematical models. When model predictions deviate from experimental data, the modeler is prompted to explore new hypotheses or increase the detail in the process descriptions, thereby guiding the selection and conducting of new experiments. Consequently, the development process is iterative, as new processes are introduced into the model and new hypotheses are examined. For example, Arav and Zohar [140] recently developed a PBPK dispersion model for the absorption of levodopa following its administration in a controlled-release formulation. The presence of dissolved levodopa in the stomach induces a ‘lag’ in gastric emptying. Incorporating this feature in the model was essential to obtaining the erratic plasma concentrations reported during the first hours after administration. Moreover, using this model, it was possible to dispute the common hypothesis of an ‘absorption window’ in the upper SI and to show that levodopa is equally absorbed along the entire SI. These results were then used to optimize the release rate of controlled-release formulations. However, because models developed under this approach require more detail, they are less suitable for high-volume screening.

First principles-based models such as molecular dynamics and continuum models offer highly detailed physicochemical representations that address processes governed by physical and chemical phenomena. These models are employed to tackle various challenges, such as comprehending the flow field induced by the contraction of the small intestine (SI) walls or optimizing the structure of nanoparticles. Due to their high complexity, both mathematically and numerically, these models are typically constrained by spatial and temporal scales. Consequently, the primary distinction between mechanism-based and first principles-based models lies in their focus. Mechanism-based models primarily operate on an organ scale or larger (e.g., multiple organs or the whole body), whereas first principles-based models concentrate on an organ scale or smaller (e.g., flow on an organ, tissue, cellular, or atomistic level).

## 8. Conclusions

In this review, we have examined the advancements in mathematical modeling of oral drug absorption. The allure of oral delivery systems coupled with the challenges and high costs associated with developing new drugs and formulations underscores the imperative of sophisticated mathematical models aimed at reducing both costs and development time.

Over the years, three primary approaches have emerged: data-driven models, mechanism-based models, and first principles-based models. These approaches differ in the problem domains they address and the mathematical tools they employ, although some overlap may occur. Specifically, data-driven models are geared towards high-throughput screening and predicting scalar values, while mechanism-based models focus on physiological problems, concentrating on dynamics at an organ scale or larger (e.g., multiple organs or the entire body). Finally, first principles-based models target physicochemical problems and zoom in to the organ scale or smaller (e.g., flow on the organ, tissue, cellular, or atomistic level).

In conclusion, all three approaches play crucial roles in enhancing oral drug delivery. A promising course of action involves integrating these approaches to leverage their respective strengths and enhance overall effectiveness.

## Figures and Tables

**Figure 1 pharmaceutics-16-00978-f001:**
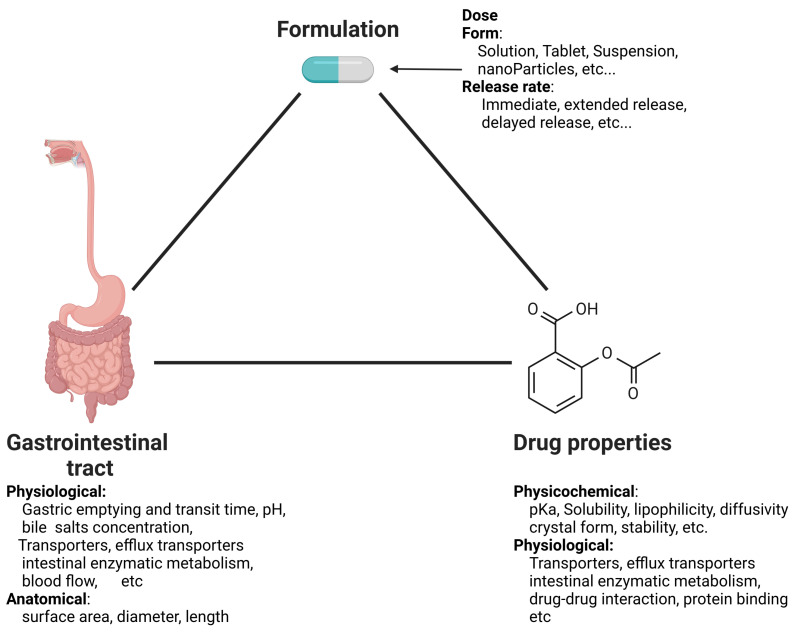
Factors influencing oral drug absorption. Created with BioRender.com.

**Figure 2 pharmaceutics-16-00978-f002:**
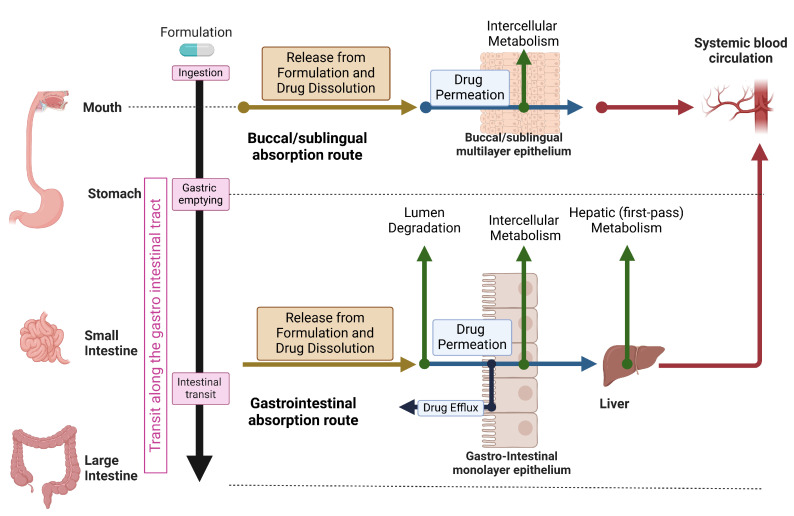
Schematic overview of oral drug absorption. Created with BioRender.com.

**Figure 3 pharmaceutics-16-00978-f003:**
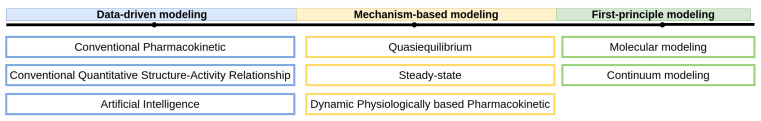
Types of models used to facilitate oral drug absorption.

**Figure 4 pharmaceutics-16-00978-f004:**
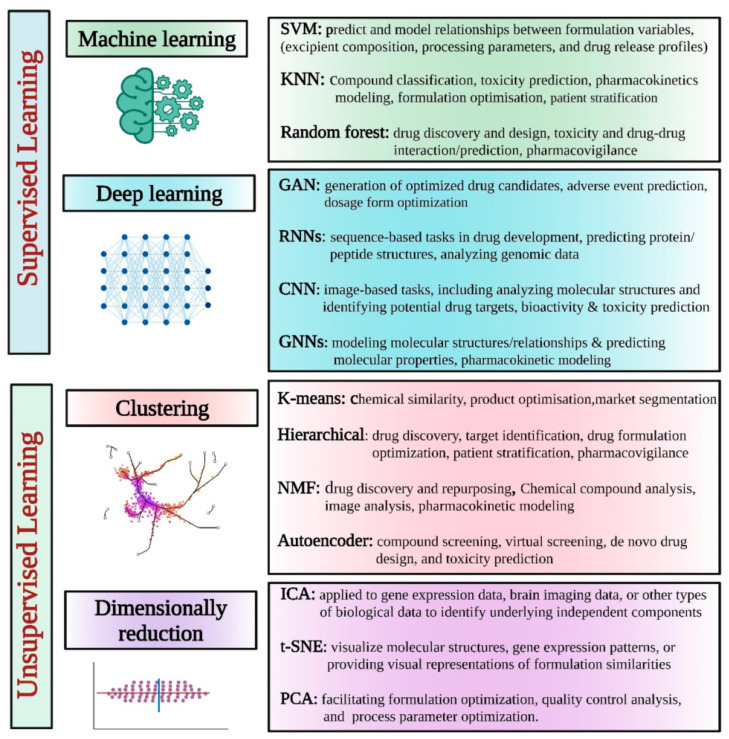
Different supervised and unsupervised AI learning models/tools for pharmaceutical applications. Taken from Vora et al. [50].

**Figure 5 pharmaceutics-16-00978-f005:**
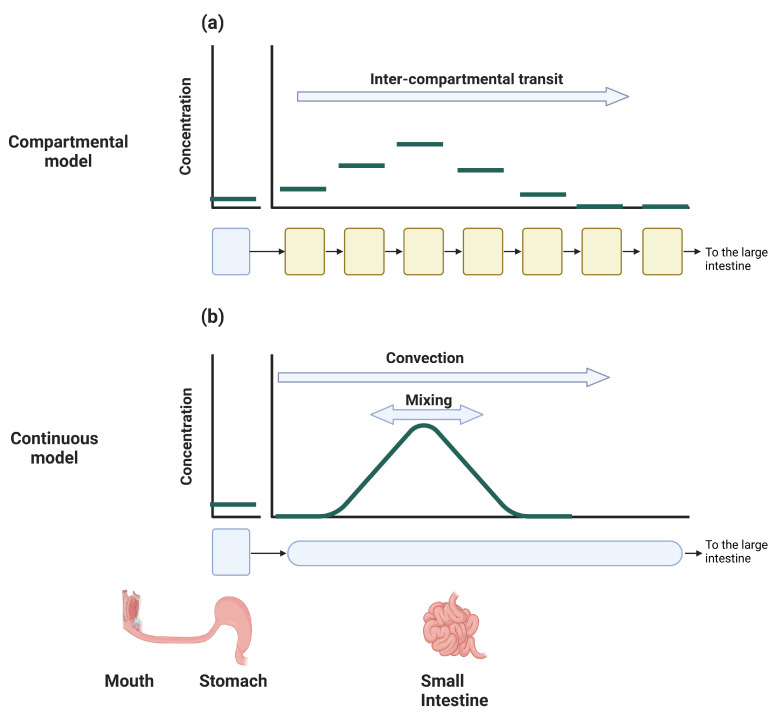
Schematic representation of the intestinal transit of dissolved drug in compartmental and continuous models: (**a**) in compartmental models, mixing is implicitly determined by the number of compartments and their assigned volumes; (**b**) in continuous models, the mixing process is explicitly described by a mixing term with a coefficient obtained from experiments.

**Table 1 pharmaceutics-16-00978-t001:** Uses and limitations of the different modeling approaches.

Modeling Approach	Usage/Properties	Limitations
Data-Driven	• High throughput screening • Extract patterns from large datasets	• Requires large datasets • Harder to provide a physical interpretation
Mechanism-based	• Focuses on physiological processes • Misprediction enhances comprehension.	• Requires physiological understanding • Results depend on the simplification methodology
First-Principles	• Focuses on physical-chemical processes. • Misprediction enhances comprehension	• High complexity limits spatial and temporal scales • Complex mathematics • Intensive computational resources

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
