# Peer review of "Advances in Modeling Approaches for Oral Drug Delivery: Artificial Intelligence, Physiologically-Based Pharmacokinetics, and First-Principles Models"

_pharmaceutics, 2024, doi:10.3390/pharmaceutics16080978_

Round 1
Reviewer 1 Report
Comments and Suggestions for Authors
In my humble opinion the manuscript entitled "Advances in Modeling Approaches for Oral Drug Delivery: Artificial Intelligence, Physiologically Based Pharmacokinetic, and First-Principal Models", submitted by Yehuda Arev, bears salient features of a review: a comprehensive and careful classification of modeling approaches based on a massive amount of references.
In fact, the author himself states in Conclusions: "In this review, we have examined the advancements in mathematical modeling of oral drug absorption."
The original contributions of the author mainly refer to the "mechanism-based" approaches and are presented in Appendices A and B! I have nothing against this (definitely original) type of presentation, provided it lies within the Journal's requirements. Otherwise, the recommendation for this manuscript would be "major revision", needed to reach a canonical form.

The author states:
"Acknowledgments: ChatGPT, a language model developed by OpenAI [? ], was used to identify and rectify grammatical errors, improve sentence construction, and enhance overall readability."
I've been able to detect some language "oddities" (some of them funny, if not hilarious) in the attached, reviewed form of the manuscript. I think a thorough "human intelligent" re-reading of the manuscript is needed. With the attitude and means of an emerging discipline/specialty/structure: counter(artificial)intelligence.
Reviewer 2 Report
Comments and Suggestions for Authors
The presented review article is devoted to the study of the introduction of machine learning into the processes of using and transporting medicines in the human body. The paper presents a summary of interesting data on machine learning, including deep learning mechanisms, basic principles and approaches used for this, and widely describes various learning models. A separate advantage is the highlighting of the specific properties of certain models and their limitations in use. I would also like to thank the author for bringing in the text of the manuscript some of the mathematical operators and formulas used in the described algorithms. The author of the review has done a lot of work on the selection of literature on this topic, many articles are extremely relevant. Among the disadvantages, I would like to note the relative brevity and superficiality of the discussion. Unfortunately, in its current form, the review, although it provides a convenient summary of the current state of the possibility of using artificial intelligence in practice to study the mechanisms of drug transportation in the human body, but the author, in fact, does not introduce any special significant novelty. Nevertheless, the work can be published as a kind of up-to-date review on the use of machine learning in the field of medicine.
Author Response
I want to express my sincere gratitude for dedicating your time and effort to reviewing my paper.
As was rightly pointed out, the paper was written as an up-to-date review of the use of machine learning in the field of medicine, among other modeling techniques.
We thank the reviewer for his remarks.
Reviewer 3 Report
Comments and Suggestions for Authors
The author proposes a mathematical model to describe the dynamics of oral drug delivery based on the detailed kinetics of the corresponding processes.
First of all, the article is under the heading "Article", but the "Abstract" section talks about "Review". The author should clarify this issue!
In the Abstract section, the author declare that three mathematical models will be considered below: firstly, data-driven models, secondly, encompassing classical pharmacokinetics, quantitative-structure models (QSAR), and, therdly, machine/deep learning.
The author pointed out that the data-driven models, also known as empirical models, are utilized to quantitatively describe empirical datasets and predict various properties. In the context of oral drug absorption, they can generally be categorized into three subcategories: Conventional Phar-
macokinetic models, Conventional Quantitative Structure-Activity Relationship (QSAR)
models, and Artificial Intelligence (AI) models that utilize Machine and Deep Learning models (ML and DL), respectively. In turn, the classical pharmacokinetic models most often describe the kinetics of absorption of a drug in blood plasma after oral administration. QSAR models establish a linear relationship, using multivariate statistics, between values of chemical descriptors computed from molecular structure and experimentally measured chemical properties, such as charge, solubility, and partition coefficient, as well as bioactivities, such as passive intestinal permeability and intrinsic hepatic clearance.AI models leverage ML and DL algorithms to augment traditional QSAR models and perform large-scale dataset classification. These models are categorized into supervised and unsupervised learning based on their learning processes. Mechanism-based models combine physiological and physicochemical principles with a simplified representation of underlying mechanisms. Developing such models presents a challenge due to the system’s complexity, involving decisions about which mechanisms to include and determining the appropriate level of detail needed for their representation.
The author pointed out that all three approaches play crucial roles in enhancing oral drug delivery. As for the review, it is a test written in a clear and understandable language and fully covers the entire range of tasks and problems in modeling of the oral drag delivery.
The review undoubtedly deserves to be published in the journal Pharmaceutics.

Author Response
I want to express my sincere gratitude for dedicating your time and effort to reviewing my paper.
Comment 1: First of all, the article is under the heading "Article", but the "Abstract" section talks about "Review". The author should clarify this issue!
Response 1: As was rightly pointed out, this paper was written as a review, and not as an original research article. I apologize for the oversight in classifying this work as an article rather than a review. It will be fixed in the final version.